# Mesenchymal Stem Cell-Based Therapy for Retinal Degenerative Diseases: Experimental Models and Clinical Trials

**DOI:** 10.3390/cells10030588

**Published:** 2021-03-07

**Authors:** Vladimir Holan, Katerina Palacka, Barbora Hermankova

**Affiliations:** 1Department of Nanotoxicology and Molecular Epidemiology, Institute of Experimental Medicine of the Czech Academy of Sciences, 14220 Prague, Czech Republic; katerina.palacka@iem.cas.cz (K.P.); barbora.hermankova@iem.cas.cz (B.H.); 2Department of Cell Biology, Faculty of Science, Charles University, 12843 Prague, Czech Republic

**Keywords:** retinal degenerative diseases, mesenchymal stem cells, stem cell therapy, experimental models, clinical trials

## Abstract

Retinal degenerative diseases, such as age-related macular degeneration, retinitis pigmentosa, diabetic retinopathy or glaucoma, represent the main causes of a decreased quality of vision or even blindness worldwide. However, despite considerable efforts, the treatment possibilities for these disorders remain very limited. A perspective is offered by cell therapy using mesenchymal stem cells (MSCs). These cells can be obtained from the bone marrow or adipose tissue of a particular patient, expanded in vitro and used as the autologous cells. MSCs possess potent immunoregulatory properties and can inhibit a harmful inflammatory reaction in the diseased retina. By the production of numerous growth and neurotrophic factors, they support the survival and growth of retinal cells. In addition, MSCs can protect retinal cells by antiapoptotic properties and could contribute to the regeneration of the diseased retina by their ability to differentiate into various cell types, including the cells of the retina. All of these properties indicate the potential of MSCs for the therapy of diseased retinas. This view is supported by the recent results of numerous experimental studies in different preclinical models. Here we provide an overview of the therapeutic properties of MSCs, and their use in experimental models of retinal diseases and in clinical trials.

## 1. Introduction

The retina is a highly specialized structure composed of several layers of morphologically and functionally different cell types. The individual layers are mutually interconnected and their primary function is to capture a light signal via the photoreceptors and to convert it into electrical impulses. These impulses are relayed to ganglion cells and then pass through the optic nerve into the visual cortex of the brain.

Individual retinal layers have an irreplaceable role in the capture and transduction of light signals. A disease or damage to any particular cell layer has a negative impact on the surrounding cell types and is reflected by the impairment of the vision. The progression of the retinal damage results in the development of retinal degenerative disorders. Although the exact etiology, causes and starting mechanisms of these diseases are mostly unknown, many factors, such as oxidative stress, light-induced damage, chemical insults, vascular defects, cytokine imbalance, damage of blood–retinal barrier and infiltration with immune cells or aging, have been suggested to contribute to the development of retinal degeneration [1,2,3]. Irrespectively of the different etiologies and various causes of retinal disorders, cumulative damage and loss of retinal cells, chronic inflammation, immune cell infiltration and enhanced cytokine secretion by immune and retinal cells represent the main pathological signs of retinal degenerative diseases, which represent the leading cause of blindness worldwide.

## 2. Retinal Degenerative Diseases

Retinal diseases are a heterogenous and multifactorial group of light-threatening disorders, which include age-related macular degeneration (AMD), retinitis pigmentosa (RP), diabetic retinopathy (DR), pediatric Stargardt’s macular dystrophy, glaucoma and many other similar forms. Although retinal diseases have various causes and different etiologies, a common characteristic is the death or dying of the specialized retinal cells and the loss of integrity of the retina or the degeneration of the photoreceptors; and this process then results in a visual impairment and ultimately in blindness.

In several of these disorders, including AMD, the earliest changes observed are caused by a loss of the cells of the retinal pigment epithelium (RPE), which play a major role in photoreceptor nutrition and in the maintenance of homeostasis. The degeneration of RPE and photoreceptors is the main cause of AMD which may have its onset in choroidal neovascularization or in accumulation of amorphous deposits. Both these causes lead to alterations in the retina and to the impairment of its functions. Early AMD is characterized by the appearance of soft drusen and pigmentary changes in the RPE, which can progress into two forms of advanced AMD—dry and wet AMD. Both of these forms result in the loss of central vision [1]. AMD is a leading cause of vision loss, affecting tens of millions of elderly people worldwide. Similarly, in DR, which is primarily caused by hyperglycemia in diabetes mellitus, the reduction in the number of pericytes at the vascular level, and the decreased number of retinal neurons and glial cells result in the interruption of retinal integrity and a progressive loss of vision. Nearly all patients with type I diabetes and over 60% of patients with type II diabetes have some degree of retinopathy after 20 years duration. DR is a leading cause of blindness in developed countries. RP is a genetic disorder of the eye which is caused by a progressive loss of the rod photoreceptor cells on the back of the eye [4]. Similarly, Stargardt’s macular dystrophy is also an inherited retinal disease that begins in childhood or adolescence and that affects the macula. On the contrary, the main risk for a group of eye diseases called glaucoma is increasing age and high pressure in the eye. The mechanism of glaucoma is believed to be a slow exit of the aqueous humor through the trabecular meshwork, which results in damage to the optic nerve and causes vision loss. In addition to these degenerative processes associated with the loss of the specialized retinal cells, local inflammation significantly contributes to the triggering and development of retinal diseases. Among the main contributors to inflammation belong the various types of infiltrating immune cells and activated microglia. Increased numbers of glial cells have been observed in the retina with the degeneration of the photoreceptors [5]. Activated microglia can contribute to the production of proinflammatory factors and to the damage of the hematoretinal barrier [6]. Furthermore, different populations of glial cells expressing genes associated with AMD (such as *VEGFA* and *HTRA1*), have been identified in the retina with this type of disease [7]. Alternatively, it has been shown that the inhibition of the microglia delays retinal degeneration in the experimental retinal vein occlusion in mice [8].

To date, the treatment options for retinal diseases have been very limited. In the advanced stages, laser photocoagulation remains the main method of treatment for DR. Other therapeutic approaches are represented by a vitrectomy or different microsurgery interventions, which involve complicated surgery and are highly invasive procedures. Recently, less invasive treatment of some forms of retinal degenerative diseases has been based on the administration of inhibitors of vascular endothelial growth factor (VEGF) or other drugs. However, these inhibitors only induce short-term effects and just slow down the progression of the disease. Therefore, the need for a safe and less invasive approach to prevent development and to treat these sight-threatening manifestations of retinal diseases is vital.

## 3. Perspectives of Cell Therapy for Retinal Diseases

Since the loss of specialized retinal cells and local inflammatory reactions are the main causes contributing to the progression of retinal degenerative diseases, the inhibition of inflammation and a support for the surviving retinal cells appear to be prospective approaches to manage these diseases. Recent studies have indicated that various types of stem cells could contribute, by paracrine effects, to the support of the survival of the residual retinal cells, and to the inhibition of inflammation [9]. A therapeutic possibility is offered by embryonic stem cells (ESCs), which can be isolated from blastocysts and which have a high differentiation potential. Another possibility is represented by the induced pluripotent stem cells (iPSCs), prepared by the reprogramming of normal adult fibroblasts or other cells. Both ESCs and iPSCs have the potential for differentiation into various retinal cell types [10,11]. However, the use of ESCs or iPSCs is limited by the possibility of immune rejection, teratogenicity and ethical restrictions in the case of ESCs. For these reasons, mesenchymal stem cells (MSCs) show great potential and could be a prospective tool for the treatment of retinal diseases. MSCs can be obtained from the bone marrow or adipose tissue of a particular patient and after separation and culturing in vitro could be used as autologous cells without the danger of immune rejection. It has been shown that after an injection of MSCs into the vitreous body, the cells can survive for a long period of time and can protect retinal ganglion cell survival or stimulate axon regeneration after optic nerve crush [12,13].

MSCs are multipotent stem cells which can be obtained relatively easily in a sufficient amount from various types of tissues and expanded in vitro for autologous application. It has been shown that MSCs retain their differentiation potential during their in vitro expansion, and that they can be differentiated into different cell types including cells expressing RPE or photoreceptor cell markers [14,15,16]. Similarly, the anti-inflammatory properties of MSCs [17,18] and their ability to support ocular surface healing [19,20,21,22,23] have been well documented. An advantage of MSCs is also their safety in use. The experiments in animal models confirmed that the subcutaneous administration of MSCs did not induce tumor growth during several months of observation [24]. Similarly, an extensive meta-analysis of studies using MSCs in over 1000 patients did not reveal a significant association between MSC treatment and the toxicity of infusions, internal organ infection, cancer or death [25].

## 4. Mesenchymal Stem Cells

### 4.1. Characteristics of MSCs

MSCs currently represent the most frequently studied type of adult stem cells. Originally, these cells were described as a population of bone marrow-derived cells that adhere to plastic and form fibrocyte-like colonies [26]. They have differentiation potential, which they retain during their in vitro expansion, as was demonstrated by their differentiation into other cell types of the mesenchymal cell line [27,28]. For therapeutic purposes, MSCs are mainly isolated from the bone marrow or adipose tissue. However, no specific marker that could characterize these cells has been identified. According to the International Society of Cellular Therapy, human MSCs are characterized by the ability to adhere to plastic surfaces in standard culture conditions, by being positive for the surface markers CD105, CD73 and CD90 and negative for hematopoietic markers CD45, CD34, CD14, CD19 and CD11b, and by their ability to differentiate into adipocytes, chondroblasts and osteoblasts [29]. It has been shown that MSCs possess potent immunmodulatory and anti-inflammatory properties, produce a number of cytokines and growth factors, and contribute to tissue healing and regeneration. The great advantage of these cells is their relatively easy isolation from the bone marrow or adipose tissue, good growth properties during their propagation in vitro and the possibility to use them as autologous (patient´s own) cells. It has also been demonstrated that MSCs from different sources (bone marrow, adipose tissue, umbilical cord blood, etc.) have similar function properties [30,31,32]. All these characteristics make them a promising candidate for the cell therapy of inflammatory and degenerative diseases.

### 4.2. Immunoregulatory and Anti-Inflammatory Properties of MSCs

The immunomodulatory properties of MSCs are mediated by multiple mechanisms including regulation by direct cell-to-cell contact, the production of various immunomodulatory molecules, the negative effects on antigen-presenting cells or the activation of regulatory T cells (Tregs). The complexity of the immunoregulatory effects of MSCs is also evident from the observation that MSCs inhibited lymphocyte proliferation induced by mitogens and alloantigens by different mechanisms [33]. In general, MSCs have potent immunosuppressive properties. It has been shown that MSCs inhibit T and B cell proliferation, the production of cytokines and activity of cytotoxic T and NK cells [17,34]. In in vivo experimental models, the administration of MSCs prolonged the survival of skin allografts in baboons [35] and mice [36], prevented the rejection of corneal allografts [37,38], decreased the incidence of graft-versus-host disease in mice and humans [39,40] attenuated septic complications [41] and suppressed the incidence and severity of autoimmune diseases [42,43]. These suppressive effects of MSCs can be mediated by multiple mechanisms. It has been shown that MSCs express numerous molecules contributing to the immunosuppression, such as indoleamine 2,3-deoxygenase (IDO), cyclooxygenase-2 (Cox-2), TNF-α stimulated gene 6 protein (TSG-6), programmed death-ligand 1 (PDL-1) or Fas-L molecule [38,44,45,46]. Furthermore, MSCs produce a number of cytokines which can negatively influence in immune reaction. It has been shown that MSCs produce transforming growth factor-β (TGF-β) and interleukin-6 (IL-6) which are the principal cytokines regulating the development of anti-inflammatory Tregs and proinflammatory Th17 cells [47,48]. The spectrum of cytokines produced by MSCs depends on the state of their activation. We have demonstrated that a cytokine environment, where MSCs reside, considerably influences their secretory and immunoregulatory potential [49]. The beneficial effects of MSCs after their systemic application in vivo are supported by the demonstration of their ability to migrate to the site of injury or inflammation and to contribute to tissue healing and regeneration [50,51,52]. In this respect we showed that mouse bone marrow-derived MSCs (BM-MSCs) administered intravenously migrated in a significantly higher number to the injured eye than into the contralateral healthy eye [53], and that adipose tissue-derived MSCs (A-MSCs) delivered intraperitoneally into transplanted mice were detected in a significantly higher amount in skin allografts than in healthy skin [54]. It has been suggested that the cytokines and chemokines produced by immune cells in the site of an injury attract MSCs to migrate to the damage site, where they participate in the attenuation of inflammation [55,56].

### 4.3. Antiapoptotic Properties of MSCs

Degenerative and inflammatory reactions in the diseased retina are regularly associated with a locally enhanced production of a variety of cytokines. These molecules can be produced either by inflammatory immune cells or by the activated cells of the retina [57]. It has been shown in vitro and in vivo that increased levels of proinflammatory cytokines can induce apoptosis of the surrounding cells [58,59]. Moreover, chronic inflammation is associated with endoplasmic reticulum stress, which also promotes the induction of apoptosis [60]. Furthermore, proinflammatory cytokines induce changes in the expression of various genes (such as *Bcl-2*, *Bax*, *p53*) associated with apoptosis. Any damage in the retina attracts the cells of the immune system which produce chemokines and cytokines, and thus potentiate inflammatory and apoptotic reactions. Therefore, the inhibition of a local inflammatory reaction and attenuation of apoptosis might be promising approaches to alleviate and inhibit the development of retinal injury. In this respect, MSCs by their immunoregulatory, anti-inflammatory and antiapoptotic properties could also be a promising therapeutic tool for developing retinal disorders [18,61]. We have recently shown that MSCs inhibit the expression of proapoptotic genes and decrease the number of apoptotic cells in the corneal explants cultured in the presence of apoptosis-inducing proinflammatory cytokines [62].

### 4.4. The Production of Growth Factors by MSCs

MSCs are potent producers of various growth and trophic factors. Some of these factors are produced by MSCs constitutively, while others are only secreted after activation with proinflammatory cytokines, mitogens or other signals. It has been suggested that the production of growth factors and their paracrine action are the main mechanisms of the therapeutic action of MSCs. Among the growth factors which are produced by MSCs and that could contribute to retinal regeneration are hepatocyte growth factor (HGF), nerve growth factor (NGF), glial cell-derived neurotrophic factor (GDNF), insulin-like growth factor-1 (IGF-1), pigment epithelium growth factor (PEGF), fibrocyte growth factor (FGF), platelet-derived growth factor (PDGF), epidermal growth factor (EGF), angiopoietin-1, erythropoietin, VEGF and TGF-β [16,57,63,64,65,66]. Some of these factors are secreted spontaneously by untreated MSCs, and their production is enhanced after stimulation with proinflammatory cytokines [16]. On the contrary, the production of some other cytokines which are produced spontaneously (such as TGF-β, HGF) is significantly decreased in the presence of proinflammatory cytokines [16]. We also showed that higher levels of some growth factors are produced by MSCs after their differentiation into cells expressing retinal cell markers [16]. The expression of genes *Ngf*, *Gdnf* and *Il-6* was enhanced in differentiated MSCs, which suggests a higher potential of differentiated MSCs for the regeneration of diseased retinal tissue. It was demonstrated that the supernatants from light-injured retina significantly promote the secretion of neurotrophic factors by MSCs and slow down the process of apoptosis in damaged retinal cells [67]. Another study showed that the secretion of neurotrophic factors by MSCs promoted the viability of photoreceptors in vitro, and supported their survival after the subretinal transplantation of MSCs in a retinal degeneration model [68]. All these observations indicate that MSCs differentiated into cells with characteristics of retinal cells have a higher secretory activity than untreated MSCs, and could have a better regenerative potential than primary MSCs.

### 4.5. The Ability of MSCs to Differentiate into Cells with Retinal Cell Characteristics

One of the characteristics of stem cells is the ability to differentiate or even transdifferentiate into different cell types. With regards to differentiation into ocular cells, relatively extensive data exist about the differentiation of MSCs and other stem cells into cornea-like cells [69,70,71,72], but the data are less abundant on the differentiation of MSCs into neurons [73] or various types of retinal cells [74,75].

The ability of different types of stem cells and MSCs to differentiate into retinal cells has been reviewed by Salehi et al. [76]. For example, MSCs isolated from rat conjunctiva and cultured in the presence of taurine expressed markers characteristic of photoreceptors and bipolar cells [75]. Taurine, together with activin A and EGF, has been used in other studies to differentiate MSCs to photoreceptors. The cells cultured in differentiation conditions for 8–10 days expressed the *Rho* and *Rlbp* genes [74]. The same authors also showed that MSCs injected into the subretinal space are able to integrate into the retina and express markers specific for photoreceptors. Other studies demonstrated that the transplantation of MSCs into the damaged retina induced the expression of markers typical for photoreceptors, bipolar and amacrine cells in grafted MSCs [77,78,79]. Several other studies also showed the differentiation of MSCs into RPE cells [79,80], which play an important role in the nourishment of photoreceptors.

In our study, to simulate the environment of the damaged retina, we cultured mouse BM-MSCs with the retinal cell extract and with supernatant from Concanavalin A-stimulated mouse spleen cells. MSCs cultured for 7 days in such conditions differentiated to cells expressing retinal cell markers such as rhodopsin, S antigen, retinaldehyde binding protein, calbindin 2, recoverin and retinal pigment epithelium 65 [16]. Interferon-γ, present in the supernatant from activated spleen cells was identified as the main factor supporting the retinal differentiation of MSCs. In addition, the differentiated MSCs produced a number of neurotrophic factors which are important for retinal regeneration. This study, and the results of other authors [78,79,80], indicate that the signals from the damaged retina induce the differentiation of MSCs into cells expressing retinal cell markers, and that the MSC differentiation is supported by cytokines produced by activated immune cells [81,82].

### 4.6. Additional Mechanisms Contributing to the Therapeutic Action of MSCs

In addition to the ability of MSCs to produce several growth, immunoregulatory or neurotrophic factors, MSCs release various types of extracellular vesicles (EVs). These particles encapsulate different functional molecules which could support the survival of cells [83,84]. For example, it has been shown that intravitreally injected EVs were as effective as MSCs in improving vision in experimental model of retinal laser injury [85]. Similarly, Mead and Tomarev [86] showed that MSC-derived exosomes protected retinal ganglion cell function in a rat optic nerve crush model.

Furthermore, mitochondrial transfer has been described as additional mechanism which MSCs can use to support anti-inflammatory conditions and cell survival [87,88]. Since mitochondrial disfunction has been proved in many retinal diseases, the mitochondrial transfer therapy might have an impact on the treatment of retinal diseases [89].

Finally, the ability of MSCs to fuse with other cell types has been documented in various models [90,91]. Therefore, the possibility of the fusion of MSCs and the cells of diseased retina cannot be excluded, and should be considered as another mechanism contributing to the therapeutic action of intraocularly administered MSCs.

## 5. The Potential of MSCs for the Treatment of Retinal Diseases

Abundant experimental data demonstrate the beneficial therapeutic effects of MSCs on retinal diseases [92,93,94]. It has been shown that MSC transplantation significantly delays retinal degeneration, supports the regeneration of RPE, cone cells and axons, and improves the survival of retinal ganglion cells. On the basis of these encouraging results, the potential to use MSCs for the treatment of retinal diseases has been proposed and tested [95,96,97,98,99]. The main mechanisms of the therapeutic action of MSCs for the treatment of retinal diseases are shown in Figure 1.

Although several questions about the clinical use of MSCs still remain unanswered, for the purpose of great interest to use stem cells for the treatment of currently incurable retinal diseases, the first clinical trials using MSCs have been initiated [100,101]. However, before the introduction of the stem cell-based therapies into clinical practice, extensive research is needed to optimize the therapeutic procedures. For this reason, experimental animal models using pharmacologically induced degeneration of the individual retinal cell types or using animals with genetically induced retinal diseases, have been introduced. The pharmacological models use the application of sodium iodate (NaIO_3_) for the destruction of RPE cells that mimic progression of macular degeneration [102,103], or the application of *N*-methyl D-aspartic acid (NMDA) or *N*-methyl-*N*-nitrosourea (MNU) that induces apoptosis and the selective degeneration of ganglion cells and photoreceptors that are processes resembling hereditary RP or glaucoma [104,105]. The studies using these preclinical experimental models support the idea that the treatment of retinal diseases with stem cells could represent the most modern and prospective approach for the treatment of currently incurable severe retinal disorders, and to improve the patient´s quality of life [106,107,108].

## 6. Possible Problems and Limitations Associated with MSC-Based Therapy

Although stem cell therapy is safe, as shown in both animal studies and clinical trials [24,25], there are still several issues that have to be taken into account before final translation of MSC-therapy from preclinical models into clinical therapy.

First at all, there is a heterogeneity of individual MSC samples, based on differences in the cell source, isolation and culture procedure. MSCs are used at different time intervals after their isolation and different doses of cells are used. It has been documented that a longer cell culture duration has an impact on MSC morphology, secretory potential and migratory properties [109,110]. Therefore, there should be an agreement about the preparation of MSCs for individual types of application.

There is still controversy about in vivo survival of in vitro cultivated MSCs. Although MSCs are considered immune privileged cells which do not express costimulatory molecules and MHC Class II molecules, in the presence of some cytokines they can express these molecules and become a target for immune cells. Without respect to these observations, numerous studies suggested a long-term survival of allogeneic or even xenogeneic MSCs in immunocompetent recipients. In contrast to these studies, Eggenhofer et al. [111] claimed that in vitro cultured MSCs are extremely short-lived and do not survive in vivo. However, there is a possibility that in immunologically privileged sites, such as those in the eye, MSCs could survive.

Another unresolved issue is the fate and immunological functions of MSCs after their transfer into the inflammatory environment of the diseased retina. There is a possibility that immunosuppressive MSCs, transferred into an environment where there are proinflammatory cytokines present, can turn out into a cell population supporting the development of aggressive proinflammatory Th17 cells.

The route of MSC administration is also very important. After the intravenous injection of MSCs, only a small proportion of the delivered cells can be found in the eye (our preliminary observations). Therefore, for the treatment of the retina, the intraocular delivery of cells appears more effective. Using experimental models, especially based on small animals, the intravitreal injection of MSCs is the most common way. However, in the healthy eye, only a few cells can be detected in the vitreous cavity, and the occurrence of side effects after such delivery of MSCs was reported [112]. Therefore, other routes of MSC application have been tested. These approaches include subretinal application [113], suprachoroidal delivery [114] and subtenon injection [115].

## 7. The Use of MSCs for the Treatment of Retinal Diseases in Experimental Models

To study the mechanisms of retinal diseases and to validate new therapeutic approaches for these diseases, various experimental models resembling different types of retinal damage have been established and tested. These models are based on pharmacological interventions which induce the degeneration of specialized retinal cells, or utilize mutant animal strains, genetically modified recipients or various mechanical damages or injuries [116]. We review here the selected experimental models that have been used to test the therapeutic potential of the various types of MSCs.

### 7.1. Experimental Models of AMD

AMD is characterized by a progressive degeneration of the RPE and photoreceptors, and this process represents the major cause of visual impairment and irreversible blindness in the elderly population. Numerous experimental models have been established to study the individual steps of AMD progression. Transgenic experimental animal models provide systems to explore the cellular and molecular mechanisms of this disease. Some advantages are offered by laser-induced models. Other approaches are based on the application of pharmacological agents inducing pathological changes in the retina. One of the well-established pharmacological models is based on the systemic or a local administration of NaIO_3_. NaIO_3_ is a chemical which selectively induces the degeneration and death of RPE cells. It was shown in vitro that the exposure of human RPE cell line ARPE-19 to NaIO_3_ induces the activation of inflammasome, changes the expression of molecules involved in the apoptosis, induces cell dysfunctions resembling conditions in AMD and finally causes RPE cell death [117,118]. In this model, human A-MSCs decreased the levels of mRNA for proapoptotic molecules and provided a rescue effect for ARPE-19 cells cultivated in the presence of NaIO_3_ [118]. In our recent study we have observed that NaIO_3_ increases the expression of genes for proinflammatory cytokines IL-1α and IL-6 or for proapoptotic Bax and p53 molecules in cultured mouse retinal explants. This increase was inhibited in the presence of mouse BM-MSCs, or by using a supernatant obtained after the cultivation of MSCs (Palacka et al., preliminary observations).

The intravenous or intraperitoneal application of NaIO_3_ in vivo causes a rapid degeneration of the RPE cells and consequent damage to the outer nuclear layer. Increased levels of mRNA for Htra-1 a C3, the genes associated with the development of AMD, were detected in the retina of the NaIO_3_-treated recipients [103,119,120]. The intravitreal or subretinal application of NaIO_3_ thus provides a suitable experimental model for study of the late phase of nonvascular AMD called geographic atrophy. The subretinal delivery of NaIO_3_ in rats causes the formation of an atrophic area characterized by the degeneration of RPE cells and photoreceptors [121]. The intravitreal administration of NaIO_3_ in rabbits after vitrectomy induced retinal atrophy and diffused outer retinal degeneration [122]. In these in vivo models, MSCs provided protection of the retinal cells from degeneration. The intravitreal injection of human A-MSCs in mice treated with NaIO_3_ protected the RPE layer, photoreceptors and other nuclear cells from the damage [123]. Gong et al. [124] showed that rat BM-MSCs transplanted into the subretinal space can differentiate into cells expressing retinal markers, and can protect the retina in the experimental models of NaIO_3_-induced retinal damage. Since the RPE cells are the first damaged cell type in the progression of AMD, the protective effect of MSCs may be a promising option for the treatment of this condition.

### 7.2. Experimental Models of DR

DR represents a common complication of diabetes which is caused by hyperglycemia and by injury in retinal microvasculature and neurons. This disorder represents one of the leading causes of blindness globally. Despite the high prevalence of DR and extensive research, the treatment options for this disease are still strongly limited. Various experimental animal models have been established for the study of treatment possibilities. These models have been generated by a selective inbreeding or genetic modifications, the feeding of a galactose diet, or by a pharmacological induction using streptozotocin. This chemical selectively damages the β cells of the pancreas, increases blood glucose level and decreases the number of ganglion cells. To date, various animal models have been used to test the possibilities of treating DR by the application of stem cells [125]. In the majority of these models, the beneficial effects of a systemic or local application of MSCs were observed.

In models of streptozotocin-induced diabetes and DR, the application of MSCs had a positive effect on the retinal architecture. For example, Ezquer et al. [126] showed that the local application of mouse A-MSCs prevented the loss of retinal ganglion cells in diabetic mice. Levels of neurotrophic factors, such as NGF, GDNF and bFGF were increased in the eyes treated with A-MSCs. Although donor A-MSCs were found integrated into the host retina, these authors did not observe the differentiation of MSCs into retinal cells. In other studies, the intravitreal administration of MSCs obtained from the human umbilical cord attenuated capillary damage in streptozotocin-induced DR and increased levels of BDNF and NGF in the treated eyes. Donor MSCs also restored the visual function measured by ERG [94,127,128]. Yang et al. [92] showed that the administration of human A-MSCs improved the integrity of the blood–retinal barrier and ameliorated DR in streptozotocin diabetic rats. Slightly enhanced levels of BDNF in the retina were also obtained after the transplantation of neural stem cells differentiated from umbilical cord MSCs, thus suggesting that this type of cells originated from MSCs may represent another suitable option for neuroprotection in DR [127]. Since MSCs isolated from mice with DR have lower proliferative abilities and higher levels of apoptosis compared to cells from healthy individuals [129,130], attempts were made to improve their therapeutic properties with the aim of using these cells for autologous transplantation in patients with DR. It has been shown that the treatment of BM-MSCs from mice with streptozotocin-induced diabetes with Wharton’s jelly extract (containing a number of growth factors and other cytokines) significantly improved their proliferative abilities and therapeutic potential [130]. It suggests that the preconditioning of diabetic MSCs could improve their therapeutic properties.

In addition to models of pharmacologically induced diabetes, several studies used models of spontaneously or genetically induced diabetes. These models were described in detail by Robinson et al. [131] and Lai and Lo [132]. For example, the Akita (*Ins2^Akita^*) mice were created by a point mutation in the *insulin-2* gene and represent a spontaneous type-l diabetes model. It was shown that hyperglycemia in these mice causes neurodegenerative effects in the retina resulting in retinal thickness [133]. In addition, elevated levels of VEGF, PEGF and placental growth factor (PlGF) and an increased expression of Iba-1 (activated glial marker) and monocyte chemoattractant protein-1 (MCP-1) were observed in the neural retina and RPE layer in Ins2^Akita^ mice during the progression of the diseases [133]. The therapeutic administration of human A-MSCs into the vitreal cavity of Ins2^Akita^ mice improved vascular permeability and vision in this model of nonproliferative DR. Similar results were also obtained after the application of the conditioned medium from human A-MSCs which were pretreated with TNF-α and IFN-γ. The conditioned medium from A-MSCs also reduced the retinal expression of GFAP, the gene associated with neuroinflammation [134]. The experimental mouse model of proliferative DR was created by the mutation causing an overexpression of VEGFa (Akimba mice). In the retina of the Akimba mice, hemorrhage and neovascularization, the degeneration of photoreceptors, the activation of microglial cells and infiltration with monocytes and macrophages were detected. The inflammatory environment is manifested by a local increase in the expression of genes for IL-1β and IL-6 and by the upregulated activation of the NLRP3 inflammasome in the retina [135,136]. Locally administered A-MSCs obtained from mice without mutation in the Insulin 2 gene were mainly found in the perivascular space and improved the vascular density in the retina [129].

### 7.3. Experimental Models for RP

RP is a group of inherited neurodegenerative diseases characterized by a loss of photoreceptor cells, leading to visual impairment and eventually to blindness. The experimental models (natural and transgenic) of this disease are based on the use of spontaneous or genetically induced degeneration of the photoreceptors, and on the administration of chemicals inducing degeneration of the retinal cells [137,138,139]. A frequently used model for RP is the rd mouse with a mutation causing the early loss of the photoreceptors. For example, the rd1 mouse is characterized by mutation in the *PDE6b* gene which is, under physiological conditions, important for the signal transmission. In addition, it was shown that activated microglia with proinflammatory polarization occur in the rd1 retina [140]. It is also possible to use an rd10 mouse which has a spontaneous mutation in the *PDE* gene for rod-phosphodiesterase. This mutation causes the degeneration of photoreceptors and other retinal cell types [141]. Moreover, it has been shown that (as with rd1 mice) activated microglia can play a role in the development of a pathological condition [142]. Another example is an rd6 mouse carrying a mutation in the *Mrfp* gene, which is expressed in the RPE layer of cells [143]. In addition to mouse experimental models, the Royal College of Surgeons (RCS) rat is often used to study RP. The RCS rat carries a mutation in the *Merkt* gene, causing photoreceptor damage and an increase in microglial activation in the retina, resulting in inherited retinal degeneration [144,145]. Some of these models have been used to study the therapeutic effects of MSCs. Treatment with MSCs has supported the survival of photoreceptors and showed therapeutic benefits. For example, the application of MSCs to the eyes of rd1 and rd10 mice provided a rescue effect for retinal cells [146]. The administration of genetically modified MSCs with an overexpression of BDNF resulted in increased antiapoptotic signaling in the retina, and in a reduction in cell damage in the rd6 mouse [147]. Moreover, the donor cells preferentially integrated into the outer retinal layers. In addition, the combined transplantation of the human retinal progenitor cells and BM-MSCs into the subretinal space provided an effective immunomodulation in the eye of RCS rats and prevented pathological changes more effectively than with a single therapy [148]. Decreased levels of TNF-α and IL-1β and an increased expression of growth factors, such as BDNF and NGF, were observed in the treated eye.

Another approach to imitate the retinal degeneration observed in RP is based on the administration of *N*-methyl-*N*-nitrosourea (MNU). A single systemic administration of MNU causes retinal degeneration in various species [139]. Deng et al. [149] showed that the treatment of mice with MNU induces retinal degeneration that can be attenuated by the administration of MSCs, and that the therapeutic effect was decreased if MSCs were prepared from the aging mice with bone progeria.

Thus, as in the case of other experimental models of retinal degeneration, the positive therapeutic effects of MSC therapy were also demonstrated in RP models.

### 7.4. Experimental Models for Glaucoma

Glaucoma is a heterogenous group of eye diseases mainly caused by increased intraocular pressure and characterized by the progressive loss of retinal ganglion cells. So far, numerous experimental models have been established to study this disease. They include the intracameral injection of microbeads, laser photocoagulation, episcleral vein cauterization, the injection of hyaluronic acid and various models based on genetically modified rodents [150,151,152,153]. These models lead to increased intraocular pressure, the degeneration of retinal ganglion cells, the activation of glial cells in the retina and increased levels of inflammatory factors in the retina [154,155]. All these models, having their advantages and limitations, were used to study new therapeutic approaches involving MSC-based therapy. Various types of MSCs have been tested with a positive impact on the decrease of intraocular pressure and on the protection of the retina. For example, Mead et al. [156] showed that an intravitreal administration of A-MSCs, BM-MSCs or dental pulp stem cells decreased ocular pressure and offered a neuroprotective effect.

A trabecular meshwork regeneration observed after intraocular administration of BM-MSCs was described in a model of a laser-induced retinal damage model [157,158]. The enhanced neuroprotective effects were observed in these models using MSCs with an increased secretion of BDNF. The neuroprotective effect of A-MSCs was also described in a model of hyaluronic acid-induced glaucoma in rats [159]. Another therapeutic approach is provided by the application of BM-MSC-derived exosomes. It has been shown that the administration of the exosomes secreted by BM-MSCs promoted the survival of retinal ganglion cells and improved the retinal structure in the eye of rats after optic nerve crush injury, and in glaucoma models with ocular pressure induced by an intracameral injection of microbeads [86]. Similarly, the application of umbilical cord MSC-derived exosomes in a glaucoma model induced by an optic nerve crush injury in rats promoted retinal ganglion cell survival and glial cell activation [160]. Thus, MSC-derived exosomes injected into the vitreous provide a significant therapeutic benefit for glaucomatous eyes and for other types of retinal degenerative diseases.

Altogether, the animal models indicated positive therapeutic effects of MSC-based therapies in various types of retinal diseases. Selected experimental models where MSCs were used are summarized in Table 1.

## 8. Clinical Trials Using MSCs for Retinal Diseases

Currently, several clinical trials are in progress to test the potential of MSCs for the treatment of retinal degenerative diseases. Most of these studies are in the phase 1 or 2 focused on the safety of MSC application. The first finished studies showed that the administration of MSCs is not associated with serious complications [164,165]. Moreover, some studies have also showed an improvement in visual function based on the examination of in visual acuity, visual field and electroretinography. The effects of the treatment have been studied in patients with variety types of retinal diseases, such as AMD, DR, RP, glaucoma, inherited retinal dystrophy, optic nerve diseases or macular holes. MSCs obtained from various sources have been used for the treatments and BM-MSC or A-MSC were tested as an option for autologous transplantation. In addition, the effects of the administration of the conditioned medium obtained from cultured MSCs or the application of exosomes prepared by the ultracentrifugation of a conditioned medium are also examined.

The application of MSCs appears safe and no serious treatment-related problems were observed in the eyes of patients with AMD, RP or retinal vascular occlusion six months after the administration of autologous BM-MSCs in the phase 1 testing. An improvement in visual function was also noted, but as the studies were designed to assess the safety of the treatment, it was not possible to definitely confirm whether this improvement was caused by the MSC application [164]. Similar results were observed in clinical trials after the application of autologous BM-MSCs in patients with RP [165]. There were no severe complications associated with cell transplantation in the treated eye. This conclusion is also supported by the study of Gu et al. [166] who showed that autologous BM-MSCs were beneficial in DR subjects with correction in macular thickness, and the improvement in visual acuity was also observed. In addition, the administration of autologous BM-MSCs improved visual acuity in patients with RP [167]. A similar improvement in visual function after the injection of autologous BM-MSC was observed in patients with optic nerve diseases and nonarteritic ischemic optic neuropathy [168,169]. In addition to the application of autologous BM-MSC, Wharton’s jelly-derived MSC transplantation improved outer retinal thickness and visual acuity in patients with RP in phase 3 clinical study [115].

However, further clinical trials with a higher number of patients and a longer follow-up are still needed to evaluate the efficacy of MSC therapy. It will also be necessary to evaluate the benefits and advantages of autologous MSCs and the transplantation of stem cells obtained from another source, such as Wharton’s jelly or the umbilical cord, or the use of a conditioned medium or exosomes. Selected clinical trials with MSCs for retinal degenerative disorders and their results are shown in Table 2.

## 9. Conclusions

Sight-threatening retinal degenerative diseases represent the main cause of visual impairment or even blindness worldwide. Despite great endeavors, there is still a lack of effective therapeutic approaches to stop or even cure these disorders. A prospective option has recently been offered by stem cell therapy. Experimental data from numerous animal models and using different types of stem cells offer promising results. Although the data from the experimental models are encouraging, numerous questions about the use of stem cell therapy have to be resolved to make this therapy more effective and safe [170]. Nevertheless, abundant clinical trials on the use of stem cells for retinal diseases have already been initiated [101,171,172]. These trials are focused on the study of the safety of the therapy, the selection of the optimal stem cells and their activation or modification prior to application, the optimalization of the dose of cells, the routes of application and the possibility of replacing the cells with their paracrine products.

One of the most important issues associated with MSC-based therapy, the safety of MSC administration, has been tested in numerous preclinical studies and clinical trials. Various experimental models and clinical studies using an intravitreal administration of MSCs demonstrated the safety of this therapy without any undesirable side effects [100,101]. Another issue that deserves attention is the mechanism of MSC therapy. Although some authors showed a long-time survival of therapeutically applied MSCs and demonstrated their presence in the eye even a few months after application, other studies suggested that MSCs are short-lived, do not survive in the recipients and can be only detected for a few days [111]. Moreover, in some experimental animal studies human MSCs were administered intravitreally and their biocompatibility, long survival and positive therapeutic effects were observed [173,174,175]. There arises a question about the mechanisms of this therapeutic effect across the interspecies barrier. Namely, Lohan et al. [176] demonstrated that human MSCs injected into rats do not have the same therapeutic effect as rat MSCs have, and that the immunoregulatory action of human MSCs is strongly limited by the interspecies barrier. These differences in the therapeutic effect between autologous/syngeneic and xenogeneic MSCs have to be taken into consideration, when human MSCs are applied therapeutically in rodent recipients and the knowledge from such experimental studies is translated to the clinical situation.

Finally, the immunoregulatory action of MSCs could strongly depend on the cytokine environment [49]. The inflammatory conditions in the diseased retina can significantly change the immunoregulatory properties of MSCs. We showed that unstimulated MSCs are immunosuppressive and spontaneously produce high levels of TGF-β, but not IL-6 [48]. TGF-β is a negative regulator of immunity and is also a factor determining the development of suppressive Tregs. However, in the presence of proinflammatory cytokines MSCs secrete, in addition to TGF-β, high levels of IL-6 [48,177]. The combination of TGF-β and IL-6 determines the development of proinflammatory Th17 cells [178]. Therefore, there is a danger that the application of MSCs into the inflammatory environment of the diseased retina could result in the inhibition of the immunosuppressive action of MSCs and in the preferential activation of proinflammatory Th17 cell population.

Although there are still many issues to be addressed before the final approval of MSC therapy for retinal degenerative diseases, the results obtained so far in preclinical animal models and in clinical trials are promising and encouraging. Therefore, the stem cell-based therapy offers a prospective option, especially for the patients without alternative therapeutic options. However, before the definitive expansion of the clinical use of MSC-based therapies, several questions, such as the sources of MSCs, the conditions of the in vitro propagation of MSCs, the routes of the cell applications or the possibility of the use of MSC products have to be answered and carefully verified.

## Figures and Tables

**Figure 1 cells-10-00588-f001:**
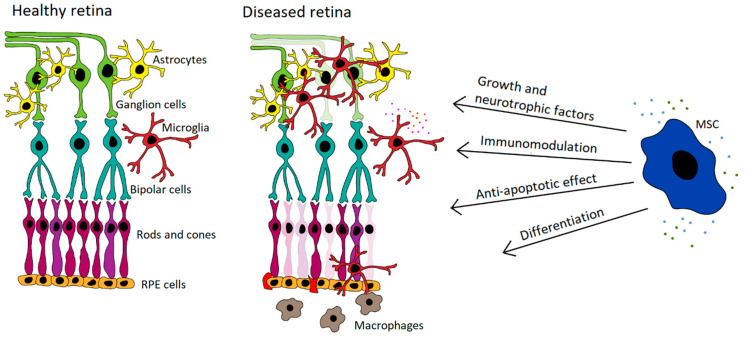
The main mechanisms of the therapeutic effect of mesenchymal stem cells (MSCs) for retinal diseases. MSCs contribute to treatment of retinal disorders by multiple mechanisms involving the production of growth and neurotrophic factors, immunomodulatory actions, by antiapoptotic effect and by direct cell differentiation.

**Table 1 cells-10-00588-t001:** Selected experimental models of MSC-based therapy for retinal degenerative disorders.

Induction of Retinal Diseases	Species	Treatment	Result	Reference
NaIO_3_	Mouse	Human A-MSCs	Protection of RPE cells, photoreceptors and outer nuclear layer	[123]
	Rat	Rat BM-MSCs	Differentiation of transplanted MSCs into cells with retinal markers	[124]
Streptozotocin	Mouse	Mouse A-MSCs	Enhanced levels of neurotrohpic factors, protection of retinal ganglion cells	[126]
	Rat	Neural stem cells (derived from humal umbilical MSCs)	Enhanced levels of neurotrohpic factors, protection of retinal ganglion cells	[127]
		Human A-MSCs	Decreased apoptosis, decrease in expression of genes related to DR	[94]
		Human umbilical MSCs	Increased expression of NGF	[128]
		Rat BM-MSCs	Improvement in visual function	[161]
Insulin 2 gene mutation	Mouse	Human A-MSCs	Decreased vascular permeability	[134]
		Conditioned medium from human A-MSCs	Decreased vascular permeability, improvement in visual function	[134]
Insulin 2/VEGFa gene mutation	Mouse	Mouse A-MSCs	Increased vascular density, incorporation of host MSCs into the retina	[129]
Cauterization of 3 episcleral veins	Rat	Rat BM-MSCs	Regulation of intraocular pressure, protection of retinal ganglion cells	[162]
Laser damage	Rat	Rat BM-MSCs	Protection of retinal ganglion cells	[157]
		Rat BM-MSCs (engineered to express BDNF)	Improvement in ERG function, protection of retinal ganglion cells	[163]
Optic nerve crush injury	Rat	Exosomes from human BM-MSCs	Protection of retinal ganglion cells	[96]
PDE gene mutation (rd 10 mouse)	Mouse	Mouse BM-MSCs	Protection of photoreceptors	[146]
Mfrp mutation (rd 6 mouse)	Mouse	Mouse BM-MSCs (engineered to express BDNF)	Induction of antiapoptotic signaling, improvement in ERG	[147]
Mertk gene mutation (RCS rats)	Rat	Human BM-MSCs with human progentitor retinal cells	Inflammatory modulation, promoting differentiation of donors cells into photoreceptor	[148]

**Table 2 cells-10-00588-t002:** Selected examples of clinical trials using MSCs for retinal degenerative diseases.

Retinal Disease	Cells For Treatment	Result	Reference
AMD, RP, retinal vascular occlusion	Autologous BM-MSC (intravitreal)	Phase 1, no severe safety issues associated with treatment	[164]
RP, cone-rod dystrophy	Autologous BM-MSC(intravitreal)	Phase 1, no severe safety issues associated with treatment	[165]
RP	Autologous BM-MSC (retrobulbar, subtenons, intravitreal, intravenous)	Improvement in visual function	[167]
Optic nerve diseases	Autologous BM-MSC (retrobulbar, subtenons, intravitreal, intravenous)	Improvement in visual function	[168]
Ischemic optic neuropathy	Autologous BM-MSC (retrobulbar, subtenons, intravitreal, intravenous)	Improvement in visual function	[169]
RP, inherited retinal dystrophy	Wharton’s jelly-derived MSC (subtenons)	Improvement in visual acuity and in outer retinal thickness	[115]
DR	Autologous MSCs (intravenous)	Improvements in macular thickness and in visual acuity	[166]
RP	Umbilical cord- derived MSC (suprachorodial)	Improvements in best corrected visual acuity, electroretinography and visual field	[114]
RP	A-MSC (subretinal)	Minor ocular complicaions, no severe safety issues associated with the treatment	[113]

## Data Availability

Not Applicable.

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
