# Peer review of "Mesenchymal Stem Cell-Based Therapy for Retinal Degenerative Diseases: Experimental Models and Clinical Trials"

_cells, 2021, doi:10.3390/cells10030588_

Round 1
Reviewer 1 Report
The article is well written and structured. The authors have done a good literature review to come up with this elaborate article, keeping mesenchymal stem cells (MSC) in focus. The article summarizes the potential of MSC for treating diseased retinas.
Author Response
Reviewer #1:
Positive evaluation, no objections or questions.

Reviewer 2 Report
Dr. Holan and colleagues’ review compile observations in experimental models and clinical trials and offer insight on potential therapeutic values in the use of mesenchymal stem cell (MSCs) in retinal diseases. The existence of numerous reviews in MSCs and their potential clinical applications including the retina diminish slightly the novelty of the paper, but nevertheless, the topic is a high relevance matter. The manuscript complies with the scope of the journal. In my opinion, it will capture the attention of basic and clinical researchers in ophthalmology and regenerative medicine fields. It is written with a proper use of the English language, it is easy to follow, and has a wide variety of references. However, there are both major and minor concerns listed in the following comments that the authors should consider before completing the final version of the manuscript.
Major points:
- The title makes the reader have the assumption that the review will describe different approaches of stem cell therapy in the retina, but the manuscript focuses only on the use of MSCs. I highly recommend the authors change the title in accordance with the content of the manuscript.
- Although the authors discuss some of the limitations and complications observed in the use of MSCs therapeutically, they do not succeed in pointing, in a clear and organized way, which are the main limitations and possible complications in the use of MSCs as a clinical therapeutic approach.
The author does not mention the heterogeneity of the isolation and culture procedures among different institutions and the consequent heterogeneity in the MSCs use in the different studies.
MSCs are considered immune privileged, but this status may not be absolute since there is some evidence of immune rejection in preclinical studies with allogeneic use.
MSCs can also have an immunosuppressive effect by generation of regulatory T cells and secretion of cytokines (Le Blanc, 2006). Proinflammatory effect has been noted in some environments (Bernardo and Fibbe, 2013; Galderisi and Giaordano, 2014). In the manuscript, the authors present these observations as a positive and beneficial outcome based on the suggestion of signalling for the migration behavior of the MSCs.
The intravitreal delivery in experimental models has raised some concerns about inflammatory and fibrous proliferation (i.e. the formation of tractional epiretinal membrane filling the vitreous cavity or even retinal detachment (Tzameret et al., 2014)).
Regarding the ability of MSCs to differentiate; the authors present this ability as a potential regenerative capacity of the therapy based on cell replacement; however, they did not contemplate the possibility of transfer. Cell replacement in the retina is currently controversial, recent observations have proved a previous misinterpretation of the identification of integrated mature photoreceptor cell after transplantation. Based on the fact that the reporter proteins used to track the transplanted cells and also endogenous proteins from the transplanted cells in the SRS have the ability to transfer to host cells (Santos-Ferreira et al., 2016; Pearson et al., 2016; Singh et al., 2016; Ortin-Martinez et al., 2017; Decembrini et al., 2017). The MSCs ability of transfer cellular material by exosomes and tunneling nanotubes has been largely demonstrated. The authors should contemplate the possibility of misinterpretations in this context also.
The weakest point in the review is the lack of a chapter dedicated to enumerating the complications observed in experimental models and clinical trials and the current limitations of the transplantation of MSCs as therapeutic approach. The inclusion of a chapter regarding the complications and limitations will substantially improve the review.
Minor points:
- Page 2, after enumerating several retinal diseases including glaucoma, the authors start the next statement “In the majority of the cases, the earliest changes observed in the retina are caused by a loss of cells of the retinal pigmented epithelium”. I agree this statement is suitable to introduce age related macular degeneration, but I cannot see how this can be correct in the case of pathologies like glaucoma or diabetic retinopathy. The authors should consider starting the statement in a different manner (i.e. In several of these disorders, including AMD, the earliest changes observed are caused by a loss of cells of the RPE).
- For the observation that the authors describe in the manuscript that are still unpublished (Kossl et al. and Palacka et al.) should be presented as preliminary observations since it has not been peer reviewed yet.
- Misspelling:
In the abstract: interconnected, irreplaceable, impairment and stress.
Page 2: worldwide and inherited
Page 3: perspectives, demonstrated and identified
Page 4: respect
Page 6: diseases
Page 7: progression, inflammasome, atrophic and intravitreal
Page 8: glucose, possibilities, umbilical, capillary, human, barrier, preconditioning, spontaneous, inflammasome
Page 9: impairment, spontaneous, decreased, offered, effect, neuroprotective
Page 10: neurotrophic, human, improvement, progenitor
Page 11: inherited, treatment
Page 12: resolved, nevertheless, issue, therapeutic
- Several typos and formatting inconsistencies remain in the manuscript.
Author Response
Dear Editors,
Thank you very much for your email of February 17, 2021, and for the invitation to revise and resubmit our manuscript ID cells-1117107 entitled „Stem Cell-Based Therapy for Retinal Degenerative Diseases: Experimental Models and Clinical Trials“ which we would like to be considered for publication in the journal CELLS.
We carefully considered all the revievers´ comments and suggestions and we made appropriate corrections and amendements. We appreciated all recommendations of the reviewers, and we believe that their suggestions contributed to the improvement of the manuscript. All changes that we made in the revised manuscript are in red font color.
Answers to the reviewers´ comments. (The order of the reviewers according to the last version):
Reviewer #2:
- As the reviewer recommended, we changed the title of the manuscript to be more in accordance with the content of the article.
- As suggested by the reviewer, we added into the revised manuscript additional chapter devoted to the limitations of stem cell therapy and possible problems which could occur in context of MSC-based therapy (new chapter No. 6).
To minor points:
- We appreciate the correction of our statement, as recommended by the reviewer. We changed our incorrect statement by a more appropriate expression suggested by the reviewer (page 2).
- As required by the reviewer, we replaced the expressions “unpublished results” by “preliminary observations” (page 8). In the case, which we reported as Kossl et al., submitted for publication, the manuscript has been accepted for publication in Stem Cells and Development. Therefore, we included this citation among References as the article in press (reference 62).
- We thanks to the reviewer for pointing misspellings. We have corrected them in the revised version of the manuscript.
- We noticed several typos and formatting inconsistencies (f.g., different size and type of letters in some words and sentences) in the printed version of our manuscript. We hope that these inconsistencies will disappear in next printed version (proofs).

Reviewer 3 Report
The article is well structured and written in proper English.
The chapters are concise and makes complex topic such as cell therapy clear and simple.
The pattern of the work is good. The synthesis on the elements to be countered in the course of retinal disease, that is, cell loss and the inflammation that is triggered, is well received. We understand that cell survival support and an anti-inflammatory effect are the therapeutic target achievable with MSC.
However, we would recommend to elaborate a few points on a more in-depth analysis to make the review more comprehensive.
To this end, I have included some suggestions in the attached file, in relation to how to complete the list and description of the properties of MSCs, introduce a paragraph that talks about the possible side effects, methods of administration and the pros and cons of these ways, deepen the paragraph on trials which appears modest compared to that on experimental models.
Conclusions need to better illustrate the future of cell therapy in light of the differences between preclinical and clinical trials.

Author Response
Dear Editors,
Thank you very much for your email of February 17, 2021, and for the invitation to revise and resubmit our manuscript ID cells-1117107 entitled „Stem Cell-Based Therapy for Retinal Degenerative Diseases: Experimental Models and Clinical Trials“ which we would like to be considered for publication in the journal CELLS.
We carefully considered all the revievers´ comments and suggestions and we made appropriate corrections and amendements. We appreciated all recommendations of the reviewers, and we believe that their suggestions contributed to the improvement of the manuscript. All changes that we made in the revised manuscript are in red font color.
Answers to the reviewers´ comments. (The order of the reviewers according to the last version):
Reviewer #3:
- Thank you to reviewer for this comment. We are aware that the replacement of missing retinal cells is probably not the major mechanism of the therapeutic effect of MSCs. The most important mechanism is a paracrine effect and the support for residual retinal cells in the damage retina. We described these mechanisms more clearly in the revised manuscript (page 3).
- As the reviewer recommended, we added into revised manuscript a chapter 4.6., where we shortly described additional mechanisms that can contribute to the therapeutic action of MSCs. These mechanisms include production of exosomes, mitochondrial transfer between cells and a possible fusion of MSCs with retinal cells. We also included references which are focused on these mechanisms, and more deeply describe them.
The reviewer suggested to include a chapter describing problems and limitations associated with MSC-based therapy. We added a brief description of these limitations in a new chapter 6, and some of them are described in conclusion.
- As concern paragraph Clinical trials, we agree with the reviewer that this paragraph deserves to be deepened. We added few information in this paragraph and in Table 2. However, in our review we intended to provide overview from MSCs to experimental models and to clinical trials. Already in the present form the review is extensive and therefore we did not go into deeper details. Especially in the case of clinical studies and clinical rials, there is a relatively large number of studies and published reviews, and therefore we, from the sake of brevity, did not provide detailed survey.
- These are very good questions from the reviewer. But, nobody at the present knows the answers and we do not want to speculate. Only additional experimental studies and clinical trials could provide the answers to these questions.

Reviewer 4 Report
The authors of the manuscript entitled “Stem Cell-Based Therapy for Retinal Degenerative Diseases: Experimental Models and Clinical Trials” have very well structured document starting with introduction and following by the description of retinal degenerative diseases and perspectives of cell therapy for these diseases. Overall, the manuscript is very well and clearly written, the authors carefully summarize the available literature.
I only have a few minor comments and remarks:
1/ Please correct the nomenclature of genes throughout the text. Gene symbols should be italicized throughout the text.
2/ Please correct typing errors in the text. Also in the version of the manuscript I received, the font size is not unified.
3/ In the paragraph 4.5. (Page 5) starting with sentence:
“The ability of different types of stem cells and MSCs to differentiate….”
It is not entirely clear here, which results were obtained from in vitro and which from in vivo experiments. Please describe it more precisely.
4/ Briefly characterize mouse and rat strains r1, r6, r10 and RCS, on which you describe the experiments in the section 6.3.
Author Response
Dear Editors,
Thank you very much for your email of February 17, 2021, and for the invitation to revise and resubmit our manuscript ID cells-1117107 entitled „Stem Cell-Based Therapy for Retinal Degenerative Diseases: Experimental Models and Clinical Trials“ which we would like to be considered for publication in the journal CELLS.
We carefully considered all the revievers´ comments and suggestions and we made appropriate corrections and amendements. We appreciated all recommendations of the reviewers, and we believe that their suggestions contributed to the improvement of the manuscript. All changes that we made in the revised manuscript are in red font color.
Answers to the reviewers´ comments. (The order of the reviewers according to the last version):
Reviewer #4:
- As recommended, we corrected the nomenclature of genes in the text. We made gene symbols in italic letters.
- We made our best to correct typing errors (as required by the reviewer). We hope that the font size will be unified in the next printed (proofs) version.
- As the reviewer recommended, we described the experiments in the paragraph more precisely to stress whether the results were obtained in vitro or in vivo.
- As the reviewer required, we briefly characterized the experimental animal strains entioned in the section 6.3 (page 10).

Round 2
Reviewer 3 Report
Page 11 Table 1
Neurotropic or neurotrophic?
Improvment or improvement?
Page 12 Table 2
For the sake of completeness, it would be nice if studies using the subretinal and suprachoroidal route of administration were also reported in Table 2.
Author Response
Dear Editors,
Thank you for your comments and suggestions.
We have corrected two misprintings, and we included, as recommended by the reviewer, two studies into Table 2. Please, if you find any mistakes in English, feel free to correct them.
Thanks.
We believe that the manuscript is now acceptable for publication.
Regards,
Vladimir Holan
